# Developing a core outcome set for periodontal trials

**Thomas J. Lamont**[1]*, **Jan E. Clarkson**[1], **David N. J. Ricketts**[1], **Peter A. Heasman**[2], **Craig R. Ramsay**[3], **Katie Gillies**[3]

1 Dundee Dental School & Hospital, University of Dundee, Dundee, United Kingdom, 2 Newcastle University School of Dental Sciences, Newcastle upon Tyne, United Kingdom, 3 Health Services Research Unit, University of Aberdeen, Aberdeen, United Kingdom

* t.lamont@dundee.ac.uk

## Abstract

### Background

There is no agreement which outcomes should be measured when investigating interventions for periodontal diseases. It is difficult to compare or combine studies with different outcomes; resulting in research wastage and uncertainty for patients and healthcare professionals.

### Objective

Develop a core outcome set (COS) relevant to key stakeholders for use in effectiveness trials investigating prevention and management of periodontal diseases.

### Methods

Mixed method study involving literature review; online Delphi Study; and face-to-face consensus meeting.

### Participants

Key stakeholders: patients, dentists, hygienist/therapists, periodontists, researchers.

### Results

The literature review identified 37 unique outcomes. Delphi round 1: 20 patients and 51 dental professional and researchers prioritised 25 and suggested an additional 11 outcomes. Delphi round 2: from the resulting 36 outcomes, 13 patients and 39 dental professionals and researchers prioritised 22 outcomes. A face-to-face consensus meeting was hosted in Dundee, Scotland by an independent chair. Eight patients and six dental professional and researchers participated. The final COS contains: Probing depths, Quality of life, Quantified levels of gingivitis, Quantified levels of plaque, Tooth loss.

**Data Availability Statement:** All relevant data are within the manuscript and its Supporting Information files.

**Funding:** TL received research funding for the core outcome set development for the prevention and management of periodontal diseases which was provided by the Tattershall fund, Dundee Dental School. This grant provided funds for the e-Delphi software, SHARE services and the face-to-face consensus meeting travel and catering costs. The funders had no role in study design, data collection and analysis, decision to publish, or preparation of the manuscript.

**Competing interests:** The authors have declared that no competing interests exist.

## Conclusions

Implementation of this COS will ensure the results of future effectiveness trials for periodontal diseases are more relevant to patients and dental professionals, reducing research wastage. This could reduce uncertainty for patients and dental professionals by ensuring the evidence used to inform their choices is meaningful to them. It could also strengthen the quality and certainty of the evidence about the relative effectiveness of interventions.

## Registration

COMET Database: http://www.comet-initiative.org/studies/details/265?result=true

## Background and objectives

Periodontal diseases are inflammatory diseases that affect the soft and hard tissues supporting teeth or 'the periodontium'. Periodontal diseases are largely preventable, yet remain one of the major causes of poor oral health worldwide and is the primary cause of tooth loss in older adults [1–4]. Periodontal diseases share common risk factors with other chronic diseases and conditions, such as obesity, heart disease, stroke, cancer, chronic obstructive pulmonary disease and diabetes [5–10].

Several interventions for the prevention and management of periodontal diseases are only supported by low quality evidence [11–14]. There is a wide variety of outcomes and clinical indices reported in trials. This outcome heterogeneity has been highlighted in guidance documents as well as Cochrane systematic reviews [13, 15–20]

It can be difficult to compare or combine studies if different outcomes are investigated and reported. This results in research wastage as these trials cannot fully contribute to improved decision making for patients and dental professionals [21, 22].

There is currently no agreement (amongst dental professionals or patients) as to which outcomes should be measured when investigating interventions for periodontal diseases.

The COMET (Core Outcome Measures in Effectiveness Trials) initiative [23] develops core outcome sets (COS), that are defined as an agreed, standardised collection of outcomes which should be measured and reported in all effectiveness trials for a specific clinical area or intervention. Core outcome sets represent the minimum that should be measured and reported upon in all trials. All relevant stakeholders should be involved in the development of a COS; it is important that patients say what outcomes matter most to them. The COS development process is concerned about what outcomes should be measured, not how these outcomes are measured.

A COS for periodontal diseases will establish through consensus a minimum list of outcomes that are relevant to patients and clinicians to be used in future effectiveness trials. This will reduce future research waste and improve care.

We aimed to develop a core outcome set for effectiveness trials investigating interventions for periodontal diseases. This COS would not be limited by health status, age or clinical setting.

## Methods

The development of this COS followed best practice and involved three stages: (1) Identification of existing outcomes; (2) Filling in gaps in knowledge and prioritisation of outcomes using Delphi survey; and (3) Face-to-face consensus meeting to finalise COS. The methods for

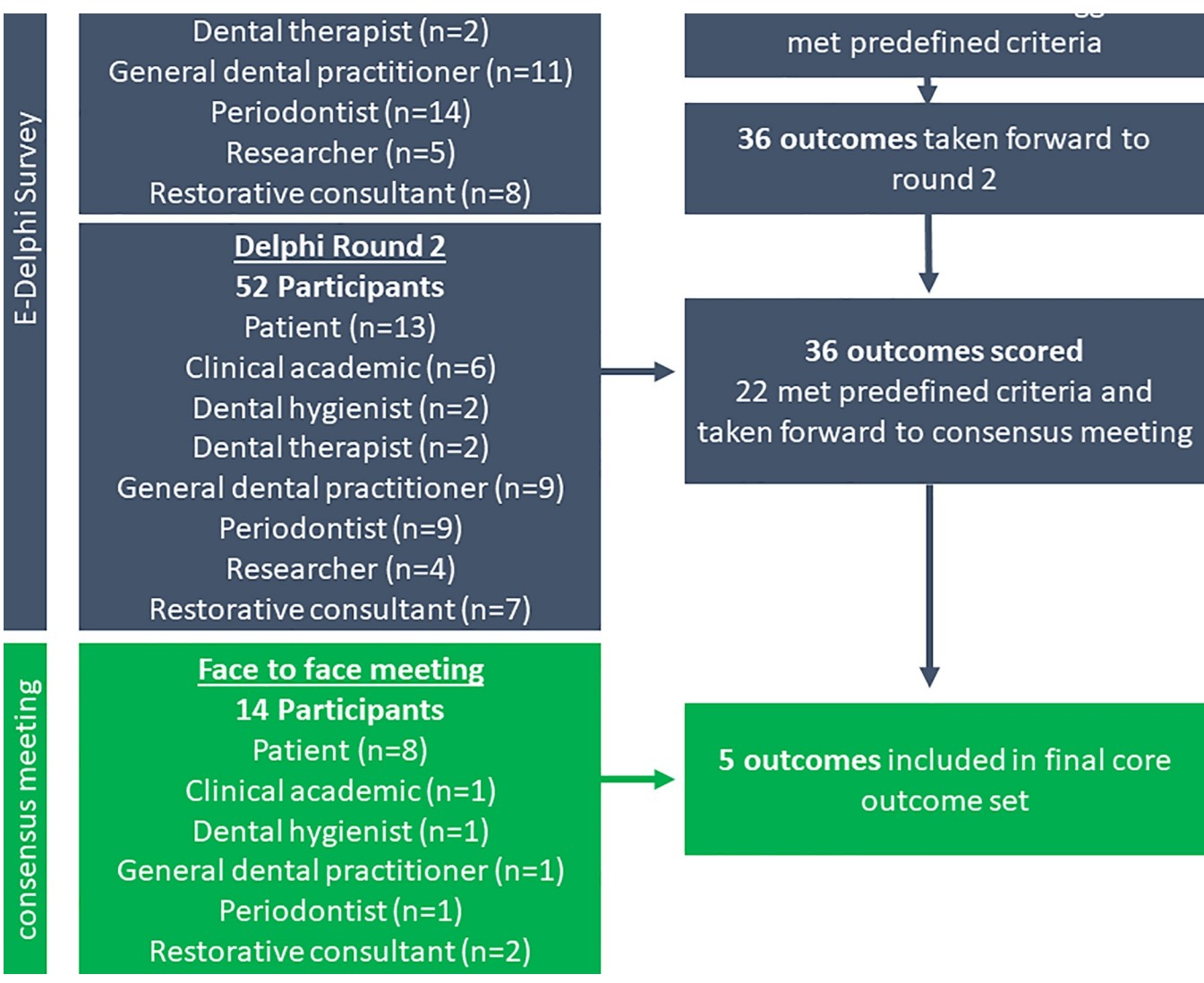

**Fig 1. Flow diagram overview of overall core outcome set development.**

each stage are outlined in detail below and in Fig 1. The study was registered on the COMET database [24]. The study protocol was developed and published in an open access peer reviewed BMC Trials journal [25]. Our report is in accordance with recommendations of the Core Outcome Set-STAndards for Reporting (COS-STAR) checklist [21].

## Ethical approval and protocol registration

Ethical approval was granted by the University of Dundee Schools of Nursing and Health Sciences and Dentistry Research Ethics Committee (Ref: 2016028_Lamont). Informed consent was obtained from all participants when they opted in to participate in the e-Delphi process. Participants of the face-to-face consensus meeting provided written consent.

## Stage 1: Identification of existing outcomes

To identify existing outcome domains in the literature we searched the Cochrane database of systematic reviews for relevant published reviews and protocols investigating the prevention

and treatment of periodontal diseases. The search was conducted up to July 2016. From the reviews and protocols that met our inclusion criteria we recorded the type of intervention(s), outcome measures (clinical, patient and economic) and duration of follow-up. We extracted data from all the trials within the included reviews and recorded any additional outcomes and indices reported by these trials that were not reported by the review. Different trials used various terminologies for the same outcome; these were de-duplicated to produce a list of unique outcomes. The outcomes were categorised as clinical; person-centred or economic. The lead investigator, Thomas James Lamont (TJL) independently extracted the data and the results were randomly reviewed by a second investigator (Jan Clarkson). Results were tabulated using Microsoft Excel (Microsoft Corporation, Redmond, WA).

## Stage 2: Filling in gaps in knowledge and prioritising outcomes

The e-Delphi process used the COMET initiative Delphi Manager [26] which is an online platform that facilitates the Delphi process.

Participants were allocated into two stakeholder groups: (1) Patients and (2) Dental professionals and researchers. There is no commonly accepted methodology for sample size calculations for e-Delphi studies [26]. The e-Delphi group size does not depend on statistical power but dynamics of reaching consensus; the literature recommends at least 10–18 per group [27, 28]. For appropriate representation we aimed to recruit approximately 20 patient participants and 50 dental professionals and researchers initially to ensure that over 50 of these participants completed the study.

Patient participant recruitment to this study was facilitated by SHARE–the Scottish Health Research Register. Inclusion criteria for patient participants: ≥18 years old, literate with access to the internet and willing to take part. There were no absolute exclusion criteria. SHARE contacted potential participants via telephone enquiries and email invitations. Potential participants were also provided with a study information sheet providing project goals and methodology (S1 Appendix).

To ensure representation of the dental profession and researchers multiple recruitment strategies were utilised for this broad stakeholder group. Dental hygienists, dental therapists, general dental practitioners, periodontists, restorative consultants, clinical academics, researchers and public health dentists were all considered potential participants. The British Society of Periodontology, Scottish Dental Practice Based Research Network and The Royal Odonto-Chirurgical Society of Scotland circulated a study invitation (including study information) to their members. The British Society of Dental Hygiene and Therapy and the Scottish Dental Practice Based Research Network advertised the study on their websites and members of the Faculty of General Dental Practitioners (Scotland) and The Glasgow Odontological Society were informed about the study.

To reduce dropout rates between rounds of the e-Delphi potential participants were asked to contact the lead researcher (TJL) to demonstrate a willingness to take part in the study. This initial stage is thought to identify those potential participants that would actively participate in the process, rather than passively participate.

**Round 1.** The outcomes were presented by domain (clinical, patient-orientated, economic) in alphabetical order. Participants were asked to score each outcome from this list using the scale proposed by the GRADE group [29], in which 1 to 3 signifies an outcome of limited importance, 4 to 6 important but not critical, and 7 to 9 critical. Participants were asked to suggest any outcomes they considered relevant but missing from the list of outcomes. A minimum of two participants had to propose an outcome for it to be included in the next round of the process. Reminders were sent to those potential participants who had recorded

an interest in the study but not completed the survey. Additional outcomes were subsequently de-duplicated and harmonized with the rest of the list prior to round 2.

Data were extracted from the DelphiManager software and analysed in Microsoft Excel. For each outcome, the percentage of participants scoring each category of "not important at all", "important but not critical" and "very important or critical" was calculated. Those participants that did not rate an outcome or chose "N/A" did not count towards the denominator in calculating the percentage of participants rating each category. Descriptive statistics were calculated for each outcome by stakeholder group.

'Stability of opinions' for stakeholder groups and individual participants were calculated as post-hoc analyses. For stakeholder groups this was assessed by computing mean stakeholder scores for each outcome between rounds; a larger number would represent a potentially important change in opinion. At the individual level mean change in scores was assessed between rounds for each individual participant across all outcomes [30].

Responses were summarised by stakeholder groups: (1) patient participants and (2) all other participants. We specified in advance that to be retained in the second round of the e-Delphi process, outcomes required 50% or more of the respondents in either stakeholder group to score it 7 to 9 and fewer than 15% score it as 1 to 3.

**Round 2.**   Participants completing round 1 were invited to round 2 and reminded of their own scores for each outcome. They were also informed of the percentage of individuals from each stakeholder group that rated scores 1 through to 9 for each outcome.

Participants were invited to rescore each outcome remaining in the e-Delphi process and score any additional outcomes that were introduced following the round 1 suggestions.

For the second round of consensus, 70% or more of the respondents in both groups had to score the remaining outcome's inclusion as critical (7 to 9) and fewer than 15% as not important (1 to 3).

## Stage 3: Consensus meeting

All participants of the e-Delphi process were invited to register their interest in attending the face-to-face consensus meeting. We employed a pragmatic recruitment approach to ensure adequate patient and dental professional participation by advertising the meeting locally via patient volunteer groups and staff emails. The inclusion criteria were the same as the e-Delphi process, namely over 18 years old and willingness to participate in the process.

The face-to-face consensus meeting utilised a modified Nominal Group Technique meeting design as the idea generation or identifying existing knowledge and fillings gaps in knowledge stages had already taken place. The priority of the meeting was finalising the recommended core outcome set.

Although definitive evidence is lacking, a review of nominal groups recommended that group sizes should remain small, no more than 14 participants, as prioritisation and consensus can be more difficult to achieve with larger groups [31, 32].

The consensus meeting was facilitated by an independent chair, Dr Katie Gillies (KG) from the Health Service Research Unit who has experience in mixed-methods research and Nominal Group Technique meetings. The meeting started with an introduction, overview of the core outcome set development process, discussion of ground rules and written consent. To encourage group discussion and sharing of views the participants were divided into two smaller groups with representation of both stakeholder groups in each. Participants were assigned to one of two tables and each group had a facilitator (KG and TJL) whose role was to clarify any issues and ensure all participants were able to voice their views.

The groups were provided with the list of outcomes from the e-Delphi process. The participants were given 60 minutes to discuss the outcomes and identify the 'top 10' outcomes they felt should be considered further. Following this first discussion stage participants were given a short break during which TJL and KG identified prioritised outcomes common to both groups. These common outcomes would be automatically taken through to the next stage. Any remaining outcomes identified by one of the groups as important would be discussed and whole group consensus of the 'top 10' outcomes established.

The groups were brought together to discuss the remaining outcomes. Participants were asked to vote yes/no if they thought the outcome in question should be included in the core outcome set for effectiveness trials investigating the prevention and management of periodontal diseases. Descriptive statistics were calculated for each of top 10 outcomes voted on by the stakeholders. The pre-specified agreement criteria for reaching consensus was 70% of all participants agreeing that an outcome should be included in the final set. The number of outcomes to be included in the final set was not pre-specified.

## Results

### Stage 1: Identification of existing outcomes

Eight Cochrane systematic reviews and three protocols were included from the 194 reviews and protocols published by Cochrane Oral Health (S1 Table). The predetermined primary and secondary outcomes of these Cochrane reviews and protocol were collated (S2 Table). The published Cochrane reviews included 134 unique studies and 23,276 unique participants. Following de-duplication of the outcomes that Cochrane review authors stated they would investigate, 25 unique outcomes were identified from the eight reviews and three protocols (S3 Table). An additional 12 outcomes were identified from the included trials of six of the published Cochrane reviews (S4 Table). The flow diagram of the identification of existing outcomes is presented in Fig 2). The long list of outcomes is presented in Table 1.

### Stage 2: Filling in gaps and prioritising outcomes

Recruitment of dental professionals commenced on the 13 November 2017 by email invitation to the British Society of Periodontology members. Patient recruitment was commenced by SHARE on the 15 November 2017. Recruitment continued until the close of round one on the 19 December 2017 at which point the recruitment target had been met.

**Round 1.** 49 potential patient participants agreed to be sent a formal invitation to the trial. A total of 61 dental professionals and researchers contacted to express interest in taking part in the study. From this, 22 patient participants and 51 dental professional and researchers registered to take part in the study; with 20 and 51 (respectively) actually completing round 1. The demographics of participants are presented in Table 2. Taken from stage 1, 37 outcomes were included in round 1 of the e-Delphi process. From this 12 outcomes were excluded as less than 50% of both stakeholder groups scored those outcomes as critical (7–9).

A total of 68 suggestions of missing outcomes were provided by 28 dental professional and research participants. Following de-duplication 11 separate outcomes met the inclusion criteria of more than two participants recommending it as a new outcome (S5 Table). Combining these 11 additional outcomes to the 25 outcomes meeting the consensus criteria of round 1 resulted in 36 unique outcomes being taken forward to round 2.

**Round 2.** Round 2 was open from the 9 to 29 January 2018. A total of 52 participants (13 patients and 39 dental professionals and researchers) completed both round 1 and round 2. 14 outcomes were excluded as less than 70% of participants scored these outcomes as critical (7–

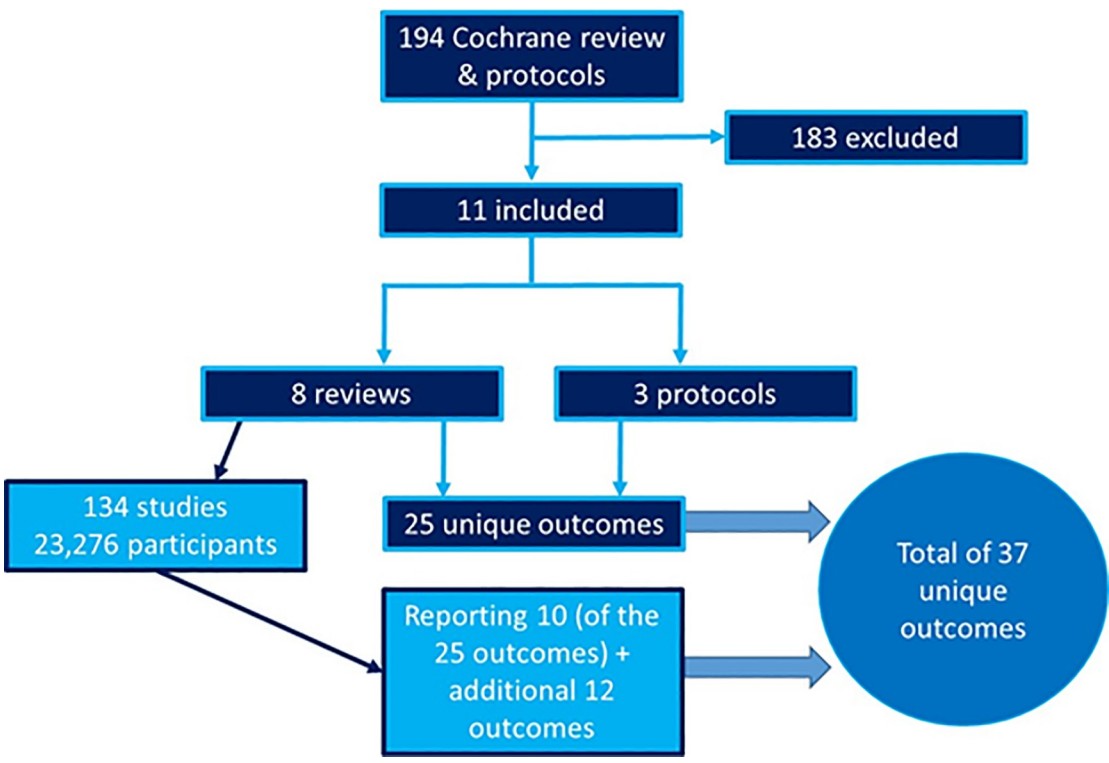

**Fig 2. Flow diagram of identification of existing outcomes.**

9). Two of the patient participants reported that they struggled to fully understand or contextualise the outcomes.

There was very little change in the mean scores of each stakeholder groups and individuals between rounds, the so called 'stability of stakeholder opinions' (S6 and S7 Tables). As the scores for each stakeholder group and individuals remained largely stable it was questionable whether the feedback had any influence at all and it was unlikely that an additional round would further improve consensus. Therefore, the Delphi was stopped after two rounds. The 22 unique outcomes taken forward to the face-to-face consensus meeting are summarised in Table 3.

## Stage 3: Consensus meeting

The meeting was hosted in Dundee Dental Education Centre, Scotland on 19 April 2018 and facilitated by KG. The lead researcher (TJL), was present throughout the meeting to provide clarification on the study purpose, scope or outcomes included.

A total of 14 participants attended (eight patients and six dental professional and researcher). Six of these participants had been involved in the e-Delphi process (one patient and five dental professionals and researchers). The participants involved in the periodontal core outcome set meeting worked well together, actively seeking and considering different opinions. The participants worked together to explain the outcomes to one another as they were discussed. It is likely that the patient participants would have struggled to fully understand outcome definitions or implications without the dental professionals' explanation. One example of this is during the discussion on the 'smoking status' outcome, when patient participant E stated: "I think we've got to be guided, as non-professionals we've got to be guided by our professionals in respect to smoking".

**Table 1. Unique outcomes identified from Cochrane reviews, protocols and included studies.**

| Outcome | Domain |
| --- | --- |
| Abrasion | Clinical |
| Body Temperature | Clinical |
| Calculus | Clinical |
| Clinical attachment loss | Clinical |
| Compliance | Clinical |
| Dental caries | Clinical |
| Dental crown failure | Clinical |
| Halitosis | Clinical |
| Incidence of periodontitis | Clinical |
| Intra-crevicular exudate | Clinical |
| Irritation of oral mucosa | Clinical |
| Microbiological parameters | Clinical |
| Nutritional status | Clinical |
| Oral infection | Clinical |
| Probing depths | Clinical |
| Quantified levels of gingivitis | Clinical |
| Quantified levels of plaque | Clinical |
| Recession | Clinical |
| Relative interdental papillary level | Clinical |
| Respiratory disease | Clinical |
| Staining | Clinical |
| Tooth loss | Clinical |
| Wear of toothbrushes | Clinical |
| Analgesics required | Patient reported |
| Average pain scores | Patient reported |
| Changes in taste perception | Patient reported |
| Patient reported behaviour change | Patient reported |
| Patient reported change in knowledge | Patient reported |
| Patient reported health | Patient reported |
| Quality of life | Patient reported |
| Reliability | Patient reported |
| Satisfaction with actual care received | Patient reported |
| Satisfaction with appearance | Patient reported |
| Satisfaction with product | Patient reported |
| Satisfaction with provider of care | Patient reported |
| Self-efficacy beliefs | Patient reported |
| Cost | Economic |

Unique outcomes and domains

The results of the prioritisation of the final 22 outcomes are summarised in Table 3. At the face-to-face consensus meeting the participants discussed the final 22 outcomes and prioritised their 'top 10' outcomes. The participants discussed these outcomes in greater detail and were asked to vote yes/no if they thought the outcome in question should be included in the core outcome set for effectiveness trials investigating the prevention and management of periodontal diseases (S8 Table). The final COS contains 5 outcomes: Probing depths, Quality of life, Quantified levels of gingivitis, Quantified levels of plaque, Tooth loss (Table 4).

**Table 2. Demographics of participants of e-Delphi study and consensus meeting.**

| Patients | Round 1 survey N = 20 | Round 2 survey N = 13 | Consensus meeting N = 14 |
|---|---|---|---|
| Male, N (%) | 9 (45) | 4 (31) | 3 (38%) |
| Age range, N (%) | | | |
| 18–30 | 0 (0) | 0 (0) | Not recorded |
| 31–45 | 5 (25) | 2 (15) | Not recorded |
| 46–70 | 14 (70) | 10 (77) | Not recorded |
| >70 | 1 (5) | 1 (8) | Not recorded |
| Residence Scotland, N (%) | 20 (100) | 13 (100) | 8 (100) |
| **Dental Professionals and researchers** | **Round 1 survey N = 51** | **Round 2 survey N = 39** | **Consensus meeting N = 14** |
| Male, N (%) | 33 (65) | 25 (64) | 3 (50) |
| Age range, N (%) | | | |
| 18–30 | 5 (10) | 5 (13) | Not recorded |
| 31–45 | 19 (37) | 16 (41) | Not recorded |
| 46–70 | 26 (51) | 18 (46) | Not recorded |
| >70 | 1 (2) | 0 (0) | Not recorded |
| Professional role | | | |
| Clinical academic | 7 (14) | 6 (15) | 1 (17) |
| Dental hygienist | 4 (8) | 2 (5) | 1 (17) |
| Dental therapist | 2 (4) | 2 (5) | 0 (0) |
| General dental practitioner | 11 (22) | 9 (23) | 1 (17) |
| Periodontist | 14 (27) | 9 (23) | 1 (17) |
| Researcher | 5 (10) | 4 (10) | 0 (0) |
| Restorative consultant | 8 (16) | 7 (18) | 2 (32) |
| Residence | | | |
| UK | 49 (96) | 38 (97) | 6 (100) |
| EU | 1 (2) | 1 (3) | 0 (0) |
| Non-EU | 1 (2) | 0 (0) | 0 (0) |

Demographics of participants in consensus study

## Discussion

This is the first core outcome set developed for periodontology that involved patients, dental professionals and researchers. Although Core Outcome Sets have been developed widely across healthcare, their development in Dentistry and Oral health is not as established. There are however a small number developed or in progress across dentistry [33–35].

The need for the standardisation of meaningful periodontal outcomes and clinical indices has been a topic of discussion in the periodontal community [16, 20, 36]. The wide variety of outcomes and clinical indices combined with their uncertain clinical significance for patients and dental professionals has been acknowledged. This study provides a set of core outcomes that have been prioritised by patients, dental professionals and researchers.

The participants rated outcomes that had previously been reported in the periodontal literature for trials investigating the prevention and management of periodontal diseases as well as those suggested by participants of this study. The online e-Delphi process was chosen to facilitate this consensus building process as this was considered the most efficient and pragmatic study design to prioritise outcomes prior to a face-to-face consensus meeting to finalise the core outcome set [26, 37, 38]. The design allowed stakeholders from a variety of settings, geographical and professional backgrounds to consider the importance of the existing outcomes and suggest missing outcomes.

**Table 3. Final 22 outcomes taken forward to consensus meeting (in alphabetical order).**

| Outcome | Dental Professionals and researchers round 2 scores | | | Patient group round 2 scores | | | Final discussion following consensus meeting voting |
|---|---|---|---|---|---|---|---|
| | 1–3 | 4–6 | 7–9 | 1–3 | 4–6 | 7–9 | |
| Abrasion | 59% | 38% | 3% | 0% | 18% | 82% | OUT |
| Bone levels on radiographic examination | 3% | 24% | 74% | 0% | 44% | 56% | OUT |
| Calculus | 3% | 46% | 51% | 0% | 25% | 75% | OUT |
| Clinical attachment loss | 0% | 10% | 90% | 0% | 11% | 89% | OUT |
| Compliance | 3% | 0% | 97% | 0% | 25% | 75% | OUT |
| Dental caries | 15% | 67% | 18% | 0% | 17% | 83% | OUT |
| Endodontic status | 8% | 63% | 29% | 0% | 13% | 88% | OUT |
| Functional occlusion | 14% | 54% | 32% | 0% | 0% | 100% | OUT |
| Furcation Involvement | 3% | 37% | 61% | 0% | 25% | 75% | OUT |
| Incidence of periodontitis | 0% | 5% | 95% | 0% | 25% | 75% | OUT |
| Intra-crevicular exudate | 5% | 62% | 32% | 0% | 11% | 89% | OUT |
| Manual dexterity | 23% | 41% | 36% | 0% | 20% | 80% | OUT |
| Oral infection | 8% | 44% | 47% | 0% | 8% | 92% | OUT |
| Probing depths | 0% | 8% | 92% | 0% | 20% | 80% | IN |
| Quality of life | 0% | 28% | 72% | 0% | 15% | 85% | IN |
| Quantified levels of gingivitis | 0% | 18% | 82% | 0% | 25% | 75% | IN |
| Quantified levels of plaque | 3% | 10% | 87% | 0% | 17% | 83% | IN |
| Recession | 5% | 61% | 34% | 0% | 27% | 73% | OUT |
| Smoking status | 3% | 10% | 87% | 0% | 30% | 70% | OUT |
| Tooth loss | 0% | 26% | 74% | 0% | 0% | 100% | IN |
| Tooth migration | 11% | 49% | 41% | 0% | 30% | 70% | OUT |
| Tooth mobility | 3% | 24% | 74% | 0% | 20% | 80% | OUT |

Breakdown of participants scores for the final 22 outcomes taken forward to the face-to-face consensus meeting.

The face-to-face consensus meeting was the first to bring together these key oral health stakeholders to discuss what outcomes they consider important for periodontal diseases. The study design was chosen to achieve consensus on what outcomes should be included in the core outcome set by facilitating group discussion and mutual clarification of opinions between the stakeholder groups [37, 39].

The scope of the study (both interventions and health area) was deliberately broad considering outcomes for all trials that investigate the prevention and management of periodontal diseases in effectiveness trials. Prevention and management strategies for periodontal diseases have wide overlap and are not commonly considered in isolation. Periodontal care routinely involves a multi-faceted approach and therefore it was considered prudent to develop one

**Table 4. Final core outcome set for effectiveness trials investigating the prevention and management of periodontal diseases.**

| |
|---|
| Probing depths |
| Quality of life |
| Quantified levels of gingivitis |
| Quantified levels of plaque |
| Tooth loss |

Outcomes included in core outcome set

single COS for prevention and management strategies due to this wide overlap [12, 40, 41]. The core outcome set was not limited by health statues, age or clinical setting.

Asking participants to suggest any outcomes missing from the initial list provided an important opportunity to capture outcomes that have not previously been reported in the Cochrane systematic review literature and in doing so identifying potential gaps in knowledge.

A strength of the study was that it allowed all participants to contribute their opinions. Throughout the Delphi process each participant rated their opinion of the importance of each outcome. The participants involved in the consensus meeting worked well together, with patient and dental professionals actively seeking and considering differing opinions. Bringing different stakeholder groups together at one face to face meeting resulted in a deeper participant understanding of the issues, allowing each participant to fully contribute.

The final COS outcomes were all considered 'critical' for inclusion by over 70% of both stakeholder groups of the e-Delphi process. All five outcomes were reported as outcomes that would have been included in the relevant Cochrane systematic reviews. The outcomes 'Probing depths', 'Quantified levels of gingivitis' and 'Quantified levels of plaque' had also been reported in 14%, 81% and 84% of the included studies presented in this outcome literature review respectively. However, 'Quality of life' and 'Tooth loss' had not been reported in the studies.

Potential limitations of the e-Delphi study are the sample size and the loss to follow. There is no commonly accepted methodology for sample size calculations for e-Delphi studies [26]. We took a pragmatic approach to selecting sample size and the number of participants (51 participants completing both rounds) is similar to other e-Delphi studies conducted whilst developing core outcome sets [42, 43]. There was also loss to follow-up in both groups, with 7 (35%) and 12 (24%) of patients and dental professionals and researchers respectively not completing the second round. With 20 patient participants at baseline a lost to follow up of 35% has the potential to cause large variation within this group across rounds, especially as some participants rated outcomes as 'Not applicable' and therefore did not contribute to the scores of that outcome. The loss to follow up is similar to other e-Dephi Studies [43, 44] and is considered one of the main drawbacks of this research design. As stated previously the stakeholder opinions were stable across the rounds and it is likely that the participants that completed both rounds were able to represent patients at large in prioritising which outcomes continue to the next phase of the COS development.

It is clear that not all of the participants fully understood the outcomes included in e-Delphi study. A study information leaflet providing information on study aims was provided to every participant and tailored to each stakeholder group. It is unclear whether this information was not sufficiently clear or whether its length put participants off reading it. Various drafts of the information sheet were produced and piloted on members of the public. The balance between too much information, which could be off-putting for some, and insufficient information would be different for various participants. All participants were advised at the start of the process that they could contact the study lead (TJL) for clarification but none of the participants did so.

The main objective of the e-Delphi was to provide participants an opportunity to recommend additional outcomes and to prioritise outcomes to be taken forward to the face to face meeting. This objective was met by our recruited participants.

Another potential limitation is the lack of anonymity during voting which may have influenced the voting. Previous studies have used anonymous voting to reduce the risk of peer pressure, voting contamination or artificial consensus [37, 45]. Hand or ballet voting as used here has previously been used successfully in other studies [31, 46].

All of the face-to-face consensus meeting participants, and the vast majority of those involved in the delphi study are based in the UK. Although this is a potential limitation, the trials in the Cochrane reviews have been published by research teams from across the world. The wider generalisability of patient perspectives to the rest of the world is unclear. A number of the professional participants are active members of the British Society of Periodontology and have been involved in international meetings and consensus exercises. It is unlikely that dental professional and researchers opinions from those involved in the study would vary largely from international colleagues.

## Conclusions

Our study reported on the robust development of a COS for use in effectiveness trials investigating prevention and management of periodontal diseases. We propose that the outcomes of: 'Probing depths', 'Quantified levels of gingivitis', 'Quantified levels of plaque', 'Quality of life' and 'Tooth loss' should be considered the minimum set of outcomes that should be reported by all effectiveness trials investigating the prevention and management of periodontal diseases. Implementation of this COS will ensure the results of future effectiveness trials for periodontal diseases are more relevant to patients, dental professionals and researchers. This could reduce uncertainty for patients and dental professionals by ensuring the evidence used to inform their choices is meaningful to them. It could also strengthen the quality and certainty of the evidence about the relative effectiveness of interventions. Future work should focus on how these outcomes should be measured in practice, as recommended by the Consensus-based standards for the selection of health Measurement Instruments (COSMIN) initiative [47].

## Supporting information

**S1 Appendix. Participant information sheet.**
(DOCX)

**S1 Table. Included Cochrane reviews and protocols.**
(DOCX)

**S2 Table. Outcomes identified in Cochrane reviews and protocols (July 2016).** Pre-specified primary and secondary outcomes of included Cochrane reviews and protocols.
(DOCX)

**S3 Table. Unique outcomes reported in Cochrane review and included trial.** Unique de-deduplicated outcomes and the review and protocol numbers that reported these outcomes. Numbers used provided in S2 Table.
(DOCX)

**S4 Table. Additional unique outcomes reported in the included trials of the Cochrane reviews.** Additional outcomes reported by trials but not included in Cochrane review. Review and protocol numbers used provided in S2 Table.
(DOCX)

**S5 Table. Additional 'missing' outcomes suggested by e-Delphi participants.**
(DOCX)

**S6 Table. Stability of opinion scores between e-Delphi round 1 and 2 for each outcome across stakeholder groups.** A measure of the average change in score between each round of the e-Delphi study.
(DOCX)

**S7 Table. Stability of opinion scores between e-Delphi round 1 and 2 for each patient participant.** A measure of the change in an individual's scoring between each round of the e-Delphi.
(DOCX)

**S8 Table. Face to face consensus voting breakdown.** Voting scores for 'top 10' outcomes with number and percentage of stakeholder groups scores presented. Consensus for inclusion was pre-determined as 70% of participants.
(DOCX)

## Acknowledgments

The authors wish to thank all patients and professionals who took part in the Delphi process and face-to-face consensus meeting. We wish to thank Jillian Sutherland, Shirley Bell, Margaret Mooney and Lorna Barnsley for helping to organise the face-to-face consensus meeting. Patient participant recruitment to this study was facilitated by SHARE–the Scottish Health Research Register. SHARE is supported by NHS Research Scotland and the Chief Scientists Office.

## Author Contributions

**Conceptualization:** Thomas J. Lamont, Jan E. Clarkson, Craig R. Ramsay.

**Formal analysis:** Thomas J. Lamont, Jan E. Clarkson, Craig R. Ramsay, Katie Gillies.

**Funding acquisition:** Thomas J. Lamont.

**Methodology:** Thomas J. Lamont, Jan E. Clarkson, Peter A. Heasman, Craig R. Ramsay, Katie Gillies.

**Project administration:** Thomas J. Lamont.

**Supervision:** Jan E. Clarkson, David N. J. Ricketts, Peter A. Heasman, Craig R. Ramsay, Katie Gillies.

**Writing – original draft:** Thomas J. Lamont, Katie Gillies.

**Writing – review & editing:** Thomas J. Lamont, Jan E. Clarkson, David N. J. Ricketts, Peter A. Heasman, Craig R. Ramsay, Katie Gillies.

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
