## [Decision Letter · Decision Letter 0]

19 Apr 2021

PONE-D-21-08793

Developing a core outcome set for periodontal trials

PLOS ONE

Dear Dr. Lamont,

Thank you for submitting your manuscript to PLOS ONE. After careful consideration, we feel that it has merit but does not fully meet PLOS ONE’s publication criteria as it currently stands. Therefore, we invite you to submit a revised version of the manuscript that addresses the points raised during the review process.

We look forward to receiving your revised manuscript.

Kind regards,

Peter Eickholz

Academic Editor

PLOS ONE

Additional Editor Comments:

Recently data have been published on the issue of end points in periodontal research that the authors at least may wich to discuss in their manuscript: e.g. Tomasi & Wennstrom 2017 and Feres et al. 2020.

Journal Requirements:

Reviewers' comments:

Reviewer's Responses to Questions

**Comments to the Author**

1. Is the manuscript technically sound, and do the data support the conclusions?

Reviewer #1: Yes

Reviewer #2: Partly

2. Has the statistical analysis been performed appropriately and rigorously? 

Reviewer #1: Yes

Reviewer #2: No

3. Have the authors made all data underlying the findings in their manuscript fully available?

Reviewer #1: Yes

Reviewer #2: No

4. Is the manuscript presented in an intelligible fashion and written in standard English?

Reviewer #1: Yes

Reviewer #2: Yes

5. Review Comments to the Author

Reviewer #1: General Comments

This is an interesting study that has significance to the field of clinical studies in periodontology. The rationale for doing the study is reasonable. Methods used are appropriate for a study of this kind. Results are clearly presented. Conclusion are reasonable.

Specific Comments

1. The sample size seems to be very small. I understand that sample size calculations were not possible but some comment on the sample size would be helpful.

2. In line with the above comment a response rate of 61 dental professionals with a final participant rate of 51 dental professionals (across all facets of dentistry) seems very low and deserves some comment. Likewise what I s the likelihood of bias in the responses of the motivated dental professionals who participated of this study?

3. I am surprised that the people in the Patient Group could understand most if not all of the “clinical” outcomes. It is not clear how this was controlled for in the study. For example how many of these people really understood the meaning or significance of “furcation involvement”, functional occlusion”, "clinical attachment loss” etc etc?

Reviewer #2: Reviewer 1: PLOS 11.04.2021

The manuscript “Developing a core outcome set for periodontal trials” aims to develop a core outcome set for use in effectiveness trials investigating prevention and management of periodontal diseases.

Overall, this work addresses a major issue in current periodontal studies: standardized reporting of the disease to make research work comparable. This study has been registered at COMET (http://www.comet-initiative.org/studies/details/265?result=true) and a detailed and understandable method paper has been published elsewhere: doi: 10.1186/s13063-017-2169-z. However, there are some major issues, that need consideration from the author.

Major Revision needed:

Abstract:

- Page 3, line 43: It is hard to understand, why Quality of Life is the core outcome, instead of oral health related Quality of Life. The selection of the variables, as well as the final decision is not comprehensible for the reader.

- Page 4, line 46-47: “Implementation of this COS will ensure the results of effectiveness trials for periodontal 47 diseases are more relevant to patients and dental professionals. , reducing research wastage.“ In general, the author should reconsider the wording: research wastage, because it implies that all research done so far is wastage.

- Page 4, line 46-47: The author should consider to explain how the research outcome leads to “…uncertainty for patients…”.

- Page 5, line 64: Is there a difference in “Quantified levels of gingivitis; Quantified levels of plaque“ and “Gingivitis; Plaque“?

Introduction:

- The introduction is well written and the author provides the reader with all the relevant information to understand the objective of the study.

- Page 6, line 80: The author cites six papers to prove outcome heterogeneity in periodontal trials. However, all six papers investigate oral health care approaches for periodontal health. None of the papers are actually performing or reviewing periodontal treatment and its outcome. Please reconsider citation. Also consider some epidemiological approaches: doi: 10.1111/jcpe.12392.

- Page 6, line 83: Please provide citation.

- Page 6, line 84-85: Why does the author only refer to interventional trials? The title addresses all periodontal trials? Please specify, whether the current work provides outcome for interventional and observational studies. The current literature does provide some agreeable outcomes for periodontal diseases, e.g. for clinical or epidemiological approaches: doi: 10.1002/JPER.18-0006 and doi: 10.1902/jop.2016.160379.

- Page 7, line 98: “This COS would not be limited by health status, age or clinical setting.” Is this a second hypothesis? Please rephrase or relocate to Results.

- Page 7, line 99-100: “Our report is in accordance with recommendations of the Core Outcome Set-STAndards for 100 Reporting (COS-STAR) checklist.“ Please relocate to M&M.

Material & Methods:

- Page 8, line 117: The author included 11 studies (review/protocol) from the Cochrane database. Why did the author limit the search to the Cochrane database? From the selected studies, only 1 study investigates outcomes after periodontal treatment. Why did the author decide to include 10 studies comparing oral health interventions and only one focusing on periodontal treatment? Inclusion criteria for literature search is not properly stated.

- Page 9, line 132: What are the exact numbers for dental professionals (n=?) and researchers (n=?)/ or refer to Table 2.

- Page 9, line 136: The author states that there were no absolute exclusion criteria. But what about language barriers?

- Page 9, line 139: Maybe the author could provide this as supplementary material?

Results:

- Page 14, line 226: The author is providing supplementary Material: Appendix A. The table lists the literature the author chose to provide sufficient outcome variables. Please check citation – where can the reader find the full citation?

- In the Methods Paper (doi: 10.1186/s13063-017-2169-z) the author states that the literature research has been conducted up to July 2016 – here the author states “The search was conducted in July 2016.” – please clarify.

I understand the struggle of conducting such a complex mixed method study involving literature review; online Delphi Study; and face-to-face consensus meeting. However, now in 2021 the literature included is at least 5 years old. I would suggest to address this as a limitation.

- Page 15, line 260: The step from round 2 to round 3 is a bit difficult to understand. For example: clinical attachment loss (CAL) were included from both dental professionals/researchers and the patient group with 90% and 89%, respectively. But in round 3, in the final discussion, CAL were excluded as final outcome. Why? CAL is a much more reliable measurement for periodontitis instead of probing depth, because it considers gum recession. Can the author provide any plausible numbers in table 3 for round 3?

- Page 16, line 269: Quantified levels of gingivitis is one of the five core outcomes. How was gingivitis classified? The aim of the current work is to provide standardized outcomes in order to prevent “research wastage” – then the core outcomes must be clearly and understandable described in order for reproduction by other authors.

Discussion

- In the first paragraph of the discussion the author should summarize the main findings of the study.

- Page 16, line 275: “There are however a small number developed or in progress across dentistry [26-28]. “ Did they reported similar outcomes? Please compare main results of your study.

- Page 17, line 289-291: What does the author mean with “The scope of the study (both interventions and health area)…“? Also “…outcomes for all trials that investigate the prevention and management of periodontal diseases in effectiveness trials“. In conclusion, the author recommends the outcomes only for interventional studies? This part is unclear in the introduction and in the title.

- The author states in the Introduction that “This COS would not be limited by health status, age or clinical setting.“. However this statement is not taken up again in the results section or in the discussion. Please clarify.

Minor revision needed:

Introduction:

- Page 6, line 77-78: Please provide citation.

- Page 6, line 78: “There is a wide variety…” instead of “There are a wide variety…”.

Results:

- Page 16, line 270: “… Quantified levels of plaque, Tooth loss (Table 4). “

6. PLOS authors have the option to publish the peer review history of their article (what does this mean?). If published, this will include your full peer review and any attached files.

Reviewer #1: No

Reviewer #2: No

---

## [Author Response · Author response to Decision Letter 0]

2 Jun 2021

We have attached a detailed response to the reviewers feedback. We thank them for their time and advice to help improve our manuscript.

---

## [Decision Letter · Decision Letter 1]

21 Jun 2021

Developing a core outcome set for periodontal trials

PONE-D-21-08793R1

Dear Dr. Lamont,

We’re pleased to inform you that your manuscript has been judged scientifically suitable for publication and will be formally accepted for publication once it meets all outstanding technical requirements.

Kind regards,

Peter Eickholz

Academic Editor

PLOS ONE

Reviewers' comments:

Reviewer's Responses to Questions

**Comments to the Author**

1. If the authors have adequately addressed your comments raised in a previous round of review and you feel that this manuscript is now acceptable for publication, you may indicate that here to bypass the “Comments to the Author” section, enter your conflict of interest statement in the “Confidential to Editor” section, and submit your "Accept" recommendation.

Reviewer #1: All comments have been addressed

Reviewer #2: All comments have been addressed

2. Is the manuscript technically sound, and do the data support the conclusions?

Reviewer #1: (No Response)

Reviewer #2: Yes

3. Has the statistical analysis been performed appropriately and rigorously? 

Reviewer #1: (No Response)

Reviewer #2: Yes

4. Have the authors made all data underlying the findings in their manuscript fully available?

Reviewer #1: (No Response)

Reviewer #2: Yes

5. Is the manuscript presented in an intelligible fashion and written in standard English?

Reviewer #1: (No Response)

Reviewer #2: Yes

6. Review Comments to the Author

Reviewer #1: (No Response)

Reviewer #2: The authors responded in an adequate manner and addressed all critiques.

I've no further comments.

7. PLOS authors have the option to publish the peer review history of their article (what does this mean?). If published, this will include your full peer review and any attached files.

Reviewer #1: No

Reviewer #2: No

---

## [Editor Report · Acceptance letter]

13 Jul 2021

PONE-D-21-08793R1 

Developing a core outcome set for periodontal trials 

Dear Dr. Lamont:

I'm pleased to inform you that your manuscript has been deemed suitable for publication in PLOS ONE. Congratulations! Your manuscript is now with our production department. 

Kind regards, 

on behalf of

Dr. Peter Eickholz 

Academic Editor

PLOS ONE